# Assessing Adhesiveness Levels in a Dysphagia Diet for Older Adults

**DOI:** 10.3390/geriatrics9020048

**Published:** 2024-04-09

**Authors:** Tae-Heon Lee, Jin-Woo Park

**Affiliations:** Department of Physical Medicine and Rehabilitation, Dongguk University Ilsan Hospital, Goyang-si 10321, Republic of Korea; taeheon320@naver.com

**Keywords:** dysphagia, diet, viscosity, texture, ageing

## Abstract

Background: Viscosity is a common focus in the diet of patients with dysphagia. However, adhesiveness is an equally significant property that can affect swallowing function, even in semi-solid foods with similar levels of viscosity. The purpose of this study was to classify the adhesiveness of food into stages and determine whether these classifications are relevant to swallowing function. Methods: This study included 30 healthy elderly adults aged 65 years. After categorizing adhesiveness into three levels based on the results of the sensory test, 30 participants were asked to swallow representative foods at each level. A videofluoroscopic swallowing study (VFSS) was performed to determine the presence or absence of pharyngeal residues based on the level of adhesiveness. The chi-square test was used to verify whether there was a difference in remnants according to the level of adhesiveness, and significance was judged with a *p*-value of less than 0.05. Results: Adhesiveness was categorized into three distinct levels: level 1 (0–4 mJ), level 2 (4–18 mJ), and level 3 (>18 mJ). Upon examining the subjects presenting with residual material, we observed three cases of level 1 adhesiveness, 16 cases of level 2, and 25 cases of level 3. The chi-square test was used to assess the statistical significance between the levels, revealing a significant difference with a *p*-value < 0.0001. Conclusions: We presented the level of adhesiveness by dividing it into three stages and proved that it is meaningful in relation to the swallowing function. Selecting and recommending foods with an appropriate level of adhesiveness will help enhance swallowing safety in older adults.

## 1. Introduction

Dysphagia is characterized by a subjective feeling of difficulty or abnormalities in swallowing [1]. The prevalence of this symptom is 20% in the general population and affects up to 50% of older adults [2]. It can severely affect patients’ quality of life, with effects including malnutrition, dehydration, aspiration pneumonia, social isolation, depression, and anxiety [3,4].

In particular, high rates of aspiration while swallowing liquids have been observed in patients with dysphagia. Dietary changes that increase bolus viscosity are a typical form of treatment effective in reducing the risk of aspiration [5,6]. Increasing bolus viscosity was undertaken for patients with stroke and showed reduced bolus penetration and a reduced risk of aspiration pneumonia [5]. However, higher viscosity is associated with an increased likelihood of pharyngeal or hypopharyngeal residues [7].

In previous studies, we confirmed that patients’ responses were different for each type of semi-solid food, even for high-viscosity foods, and confirmed that food texture could be related [8]. Food texture was analyzed in terms of hardness, adhesiveness, and cohesiveness. Adhesiveness refers to the effort required to detach food that sticks to the mouth’s surfaces, typically the palate, during the natural swallowing process, while cohesiveness describes the internal bond strength that constitutes the food’s structure [8]. It was thought that swallowing would be easier with higher cohesiveness and lower adhesiveness; however, only adhesiveness was statistically significant. This research is particularly valuable, as it underscores adhesiveness as an important property, alongside viscosity, for older adults dealing with dysphagia.

The purpose of this study was to divide the adhesiveness of food into stages and determine if there was a difference in participants’ responses according to the stage. By analyzing the responses, it will be possible to recommend appropriate food types for older adults.

## 2. Materials and Methods

### 2.1. Subjects

Thirty healthy older adults (19 male and 11 female) with a mean age of 77.0 ± 6.5 years (range, 65–90) were recruited from the community using poster advertisements. They had no underlying conditions that could cause dysphagia and were not taking any medications that could affect their swallowing function. This study was approved by the Institutional Review Board of the University Hospital. Informed consent was obtained from all the participants.

### 2.2. Determination of Adhesiveness Level

In our previous study [8], we conducted a comprehensive evaluation of nine different foods—whipped cream, mayonnaise, soft tofu, mango pudding, boiled mashed pumpkin, boiled mashed potatoes, boiled mashed sweet potatoes, red bean paste, and peanut butter. These foods, although similar in viscosities as determined by fork test [9] grade 1, International Dysphagia Diet Standardization Initiative [IDDSI] [10] level 4 category, and British Dietetic Association [11] Texture C, exhibited varying levels of adhesiveness.

Initially, an instrumental assay was conducted to quantitatively assess the adhesiveness of each food item. Utilizing a CT3 texture analyzer (AMETEK Brookfield, Middleboro, MA, USA), we measured the adhesiveness under specific conditions—central temperature at 20 ± 0.2 °C, 70% strain, a 20 mm probe diameter, a 10.5 mm infiltration depth, and a testing speed of 10 mm/s. Adhesiveness was defined by the effort required to detach the food from the mouth’s surface, particularly quantified by the negative area representing the work needed to pull the probe away after the first bite [12]. This adhesiveness data was not re-measured in this current study but was derived from the analyses previously conducted.

Subsequently, a sensory evaluation was performed to gauge the ease of swallowing these foods based on their adhesiveness levels. Participants were presented with foods of each adhesiveness category, and their responses were charted, illustrating the perceptual ease of swallowing. Foods with adhesiveness below 4 mJ were generally found to be easier to swallow, whereas those beyond 18 mJ posed difficulty for the majority. Based on the completed graph, adhesiveness was categorized into three main levels.

### 2.3. Videofluoroscopic Swallowing Study (VFSS)

Thirty participants were asked to swallow representative foods at each level (Level 1 was represented by Ottogi mayonnaise, Level 2 by Pulmuone mashed boiled pumpkin, and Level 3 by No Brand peanut butter) and rinse their mouths once before swallowing another food. All participants underwent VFSS using a fluoroscope (Sonialvision-100, Shimadzu Corporation, Kyoto, Japan) according to the standard protocol. A uniform amount of contrast agent was mixed into each representative food, which was at room temperature, immediately before administration. Upon readiness, participants were provided with a 5 cc spoonful of the mixture and were instructed to swallow the food without mastication. This protocol was conducted in accordance with the sensory test protocols outlined in our previous study [8]. However, the 30 participants were not the same individuals who had participated in the previous sensory test. Foods with the same adhesiveness level were provided twice, and images were captured for each instance. Videofluoroscopic images were captured directly using INFINITT PACS M6 (INFINITT Healthcare Co. Ltd., Seoul, Korea). In general, when remnants are found in the pyriform sinuses or valleculae, they are quantified using a grading system of grade 1, 2, or 3. However, our experiment focused on a binary outcome (present or absent), and, therefore, we did not include such a detailed grading system in our study. Instead, to focus on the comparative evaluation of the adhesiveness levels among foods, the two physicians reached a consensus regarding the presence or absence of food remnants. The results were recorded based exclusively on the presence or absence of remnants in the oropharyngeal cavity, which could be discerned visually.

### 2.4. Statistical Analysis

All statistical analyses were performed using SPSS (version 12.0; SPSS Inc., Chicago, IL, USA). The chi-square test was used to verify whether there was a difference in remnants according to the level of adhesiveness, and significance was judged with a *p*-value of less than 0.05.

## 3. Results

Participants chose either “easy” or “hard” for each individual food item. For adhesiveness levels below 4 mJ, the majority of the participants responded with “easy”, while for levels beyond 18 mJ, the majority indicated it as “difficult”, as shown in Figure 1. Based on the participants’ responses, we stratified the adhesiveness into three distinct levels. Level 1 represented adhesiveness from 0 mJ to 4 mJ, Level 2 ranged from 4 mJ to 18 mJ, and Level 3 encompassed values above 18 mJ. Level 1 represented mayonnaise, level 2 represented mashed boiled pumpkin, and level 3 represented peanut butter. Our observations revealed remnants in 3 people in Level 1, 16 in Level 2, and 25 in Level 3, shown in Table 1. As the adhesiveness level increased, a greater number of remnants remained. When tested using the chi-square test, a significant difference was found, with a *p*-value < 0.0001.

## 4. Discussion and Implications for Future Research

The world is rapidly transitioning into an aging society, and the increase in the older population is equally rapid. The capacity to swallow, which is often taken for granted, deteriorates with age, increasing the chances of developing disordered swallowing [9]. Therefore, the interest in dysphagia is growing. In addition to aging, other factors can cause dysphagia [10].

Viscosity is the primary target of dietary modification in dysphagia management [5]. However, even with the same viscosity, textural properties such as hardness, adhesiveness, and cohesiveness vary among different foods. Through our previous research, we discovered that only adhesiveness is the property that impacts pharyngeal swallowing [8]. While there have been studies exploring the relationship between cohesiveness and dysphagia [11], they differed from our research, as they were limited by variations in viscosity among different food items.

The realization that only adhesiveness was associated with the sensation of difficulty in swallowing led us to contemplate how to regulate this sensation in a way that could benefit older adults. The first step in this endeavor was to classify adhesiveness into three distinct levels. Through our experiments, we discovered that as the adhesiveness level increased, the remnants also increased. Based on this, recommending foods with the same viscosity and adhesiveness as level 1, where the least remnants were observed, could be significantly safer for older adults by reducing the residue in the pharyngeal area and the risk of aspiration.

Since we have experimentally confirmed that remnants increase with increasing adhesiveness levels, even in healthy older adults without neurological symptoms, studies related to adhesiveness should be conducted in dysphagic patients who have underlying neurological causes as the target group. Conducting additional research to differentiate adhesiveness levels across a broader spectrum of solid-textured foods and examining their correlation with swallowing would hold meaningful value within the scope of this study.

This study revealed a stepwise correlation between adhesiveness and swallowing function. The objective confirmation of remnants at each level through VFSS imaging rather than subjective sensory tests is also a strength of this study.

This study has limitations, such as a small sample size and a focus on healthy adults over 65 years of age. Conducting research on patients with stroke or individuals affected by conditions such as Parkinson’s disease could allow for a more comprehensive discussion of dysphagia diets. Additionally, if an assessment of swallowing function, such as measuring IOPI (Iowa Oral Performance Instrument) values in volunteers or conducting preliminary VFS, had been performed in advance, the participation criteria would have been more clearly defined. Another limitation is the difficulty of realistically controlling food homogeneity (mixing saliva and temperature) in swallowing-related research.

In future research, it would be beneficial to extend the application of this research to populations with specific health conditions, such as those who have suffered a stroke or are living with Parkinson’s disease. These conditions profoundly affect swallowing mechanics, and understanding the role of adhesiveness in such contexts could yield crucial insights for dietary management in dysphagia. Additionally, incorporating a comprehensive evaluation of tongue strength, for instance, by measuring IOPI (Iowa Oral Performance Instrument) values, could enrich the participant selection criteria and provide a more detailed understanding of the swallowing function.

## 5. Conclusions

We determined the level of adhesiveness by dividing it into three stages and proved that it is meaningful in relation to swallowing. Selecting and recommending foods with appropriate levels of adhesiveness will help enhance swallowing safety in older adults.

## Figures and Tables

**Figure 1 geriatrics-09-00048-f001:**
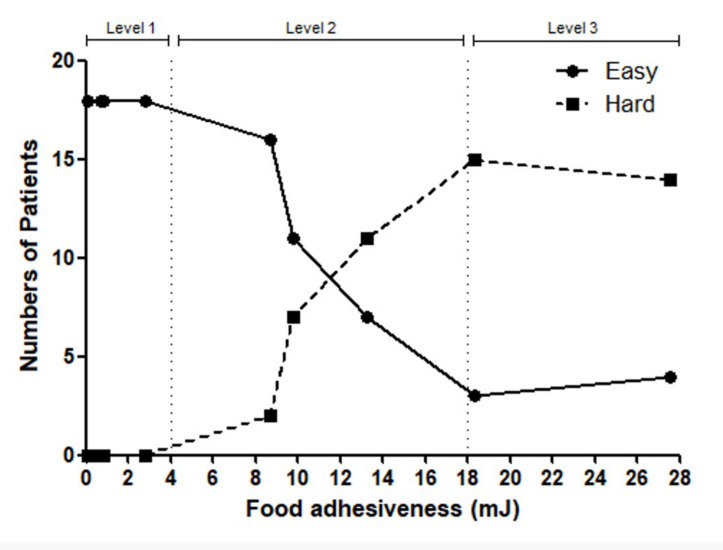
Participants chose either “easy” or “hard” for each individual food item. For adhesiveness levels below 4 mJ, the majority of participants responded with “easy”, while for levels beyond 18 mJ, the majority indicated it as “difficult”.

**Table 1 geriatrics-09-00048-t001:** Food remnants according to adhesiveness level.

		Absence of Remnants (Person)	Presence of Remnants (Person)	*p*-Value
Adhesiveness	Level 1	27	3	
	Level 2	14	16	
	Level 3	5	25	0.0001 *

* Chi-square test.

## Data Availability

All data generated or analyzed during this study are included in this published article.

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
