# Peer review of "Assessing Adhesiveness Levels in a Dysphagia Diet for Older Adults"

_geriatrics, 2024, doi:10.3390/geriatrics9020048_

Round 1

Reviewer 1 Report

Comments and Suggestions for Authors

This is a useful study, I found it interesting to read. However there are some areas of the manuscript that need more detail. Also I am concerned about the application of IDDSI and need a greater focus on how the consistency of the foods was assessed.

Introduction - explain the meaning adhesiveness and cohesiveness in associated with the sentence at the bottom of page 1. It should be in the introduction instead of the discussion.

Methods - were the participants assessed as not having any swallowing dysfunction?

Methods - how was adhesiveness measured in order to give the values in mJ in Figure 1? You've given links to the viscosity definition but not method for adhesiveness. As adhesiveness is the fundamental point in this paper, full methods should be detailed for the measurement. What were the foods used to create Figure 1? The last sentence of this paragraph is the same as in the results, and should be deleted from the methods. Move Fig. 1 to the results.

Methods - IDDSI is not a simple categorisation of viscosity, it includes such as stickiness/adhesiveness, e.g. the spoon tilt test requires the food to slide off the spoon without sticking to be IDDSI 4 (https://iddsi.org/Testing-Methods). As such I question whether the peanut butter was really IDDSI level 4. Detail about the food products used is needed - e.g. manufacturer, components.

Methods - how was the contrast agent incorporated into the food? What was the quantity or concentration of agent in the food? What quantity of food was given? What temperature was the food when provided to the participants? Were they allowed to chew or move it around to prepare for swallowing? How long were they allowed to hold it in their mouth and prepare it for swallowing before actually swallowing - was any of this planned or monitored - explain exactly what happened, whether participants were allowed to do whatever they wanted to it in their mouth. Citation for the standard protocol - has it been used in a research paper before in exactly the same way? What area was assessed for presence or absence of food remnants - just the oropharangeal cavity or how far down the oesophageal tract was included?

Results. Figure 1 is part of the results, not methods, it should be moved and cited in the first sentence of the results. The results from the assessments of the foods using IDDSI would be helpful to prove they are all IDDSI level 4 - did you do the spoon tilt test? This will help to make the paper feel like there is more to it than just three numbers and one chi-square test.

Discussion - another limitation is your lack of assessment of swallowing function of your volunteers. Would measurement of tongue strength have helped differentiate reasons for residue?

Discussion - seems to be very limited attempt to relate the work to the literature. 

Author Response

We will now begin our response to your review. Above all, we would like to thank you for reviewing our paper so carefully.

Introduction - explain the meaning adhesiveness and cohesiveness in associated with the sentence at the bottom of page 1. It should be in the introduction instead of the discussion.

  • Thank you for helping to enhance the structure of the introduction in our manuscript. As you suggested, we have incorporated the definitions of adhesiveness and cohesiveness into the introduction.

Methods - were the participants assessed as not having any swallowing dysfunction?

  • Thank you for your important observation. Due to practical constraints, such as the necessity for participants to visit the hospital for scheduled assessments, we were unable to conduct preliminary evaluations. However, we clearly stated during the recruitment phase that individuals who 'do not complain of dysphagia' were eligible for the study. Additionally, we have considered your review and incorporated relevant discussions into the manuscript to address this matter.

Methods - how was adhesiveness measured in order to give the values in mJ in Figure 1? You've given links to the viscosity definition but not method for adhesiveness. As adhesiveness is the fundamental point in this paper, full methods should be detailed for the measurement. What were the foods used to create Figure 1? The last sentence of this paragraph is the same as in the results, and should be deleted from the methods. Move Fig. 1 to the results.

  • We are grateful for the careful attention you have given to our manuscript. The sensory test was conducted based on a precedent study, a copy of which will be attached for your reference; reading it may clarify the flow of our research. In that study, a CT3 texture analyzer was utilized for texture profile analysis, and adhesiveness (mJ) was measured. We have added related details in the 'Determination of Adhesiveness Level' section of our methods. The three foods tested in the current study were prepared identically to those in the sensory test. As you recommended, Fig. 1 has been moved to the results section, and the last sentence of the paragraph has been deleted. These revisions have enhanced the clarity of our work. Thank you once again..

Methods - IDDSI is not a simple categorisation of viscosity, it includes such as stickiness/adhesiveness, e.g. the spoon tilt test requires the food to slide off the spoon without sticking to be IDDSI 4 (https://iddsi.org/Testing-Methods). As such I question whether the peanut butter was really IDDSI level 4. Detail about the food products used is needed - e.g. manufacturer, components.

  • I understand your questions regarding the IDDSI level settings. It was indeed an aspect we deliberated upon. Initially, we focused on classifying the stages based on flow tests related to adhesiveness. After all, what we wanted to determine was how such stickiness of the foods could be associated with adhesiveness. Moreover, the "Creamy peanut butter" we used, manufactured by Yantai Xufeng Food Company, was chosen because it is one of the most fluid and least sticky among commercially available peanut butters—a product that exhibits flow when lifted with a spoon or fork. Should there still be discrepancies with additional tests such as the spoon tilt test, we will retain only the fork test grade and remove the other classifications.

Methods - how was the contrast agent incorporated into the food? What was the quantity or concentration of agent in the food? What quantity of food was given? What temperature was the food when provided to the participants? Were they allowed to chew or move it around to prepare for swallowing? How long were they allowed to hold it in their mouth and prepare it for swallowing before actually swallowing - was any of this planned or monitored - explain exactly what happened, whether participants were allowed to do whatever they wanted to it in their mouth. Citation for the standard protocol - has it been used in a research paper before in exactly the same way? What area was assessed for presence or absence of food remnants - just the oropharangeal cavity or how far down the oesophageal tract was included?

  • Thank you for your detailed review of the Methods section. As mentioned regarding the measurement of adhesiveness, the sensory test was already completed in previous research. For this study, we selected three of the nine foods from that test to represent each adhesiveness level. The preparation of the food and the swallowing procedure were identical to those used during the previous sensory test. We have addressed your comments by providing additional details in the Methods section under the VFSS part. The measurement area for remnants is the oropharyngeal area.

Results. Figure 1 is part of the results, not methods, it should be moved and cited in the first sentence of the results. The results from the assessments of the foods using IDDSI would be helpful to prove they are all IDDSI level 4 - did you do the spoon tilt test? This will help to make the paper feel like there is more to it than just three numbers and one chi-square test.

  • As recommended, we have revised the flow of the results section. Regarding IDDSI level 4, as previously explained, we hope you understand that our focus was on the flow test. Although we were unable to conduct the spoon tilt test, we aimed to compensate for this by focusing on adhesiveness. We hope you can appreciate our concerns and the direction of our research once again.

Discussion - another limitation is your lack of assessment of swallowing function of your volunteers. Would measurement of tongue strength have helped differentiate reasons for residue?

  • Thank you for your observation. As previously mentioned, due to practical reasons, we were unable to conduct preliminary assessments, and we have addressed this point as a limitation at the end of the manuscript.

Discussion - seems to be very limited attempt to relate the work to the literature. 

  • Thank you for your meaningful observation. In the discussion section, we initially referenced existing literature that focused on viscosity and cohesiveness in relation to dysphagia. Our research centered on adhesiveness, a topic not extensively covered in the mentioned studies. Given the abundance of research on viscosity, there were limitations in establishing a direct relation to existing papers. We believe that as research on adhesiveness continues to advance, these limitations can be addressed.

Reviewer 2 Report

Comments and Suggestions for Authors

Authors propose a simple but straight assay to demonstrate the relevance of perceived adhesiveness on the swallowing function. This kind of research actions are required to include other key parameters within the dietetic management of dysphagia, which for many years was focused on viscosity. I think that this contribution is relevant on that direction. However, methodology part should be further explained so that it gives practical information to be used in diets for dysphagia. Offering both viscosity and adhesiveness values for the food samples used, together with swallowing ease or difficulty would constitute a relevant piece of information for dysphagia management.

I would recommend performing major changes before considering this manuscript to be published in Geriatrics.

Some aspects to address are the following ones:

INTRODUCTION

Lacks background on adhesiveness

MATERIALS AND METHODS

- Express age values as mean ± SD in materials and methods

- The sensory test Fork test 1 and IDDSI level 4 and British Classification C are treated as equivalent or similar regarding viscosity. However, Fork test 1 refers to thickened liquids which don´t drip from the fork. On the contrary, IDDSI level 4 and British C level refer to pureed smooth food. It is noteworthy that IDDSI level 4 is also described as “not being sticky” in the DDSI Spoon Tilt Test. This particularity is not described in the Fork test by Park J-W and should be taken into account. On the other hand, Texture C by the British classification refers to thick purée and it can be explained by the following characteristic: “the prongs of a fork make a clear pattern on the surface” (which it might not happen in cat 1 of the fork test), together with the fact that it should not be sticky, as in IDDSI level 4.

I would say that it is too general to describe the range of viscosities tested this way, just mentioning classification categories, and that it is necessary to clearly report which were the foodstuffs used and which were their viscosity values. Sometimes it might be required that author also explained how foodstuffs were prepared.

- Please, describe in details of the design and experimental conditions of the sensory test performed to define levels of adhesiveness.

In the sensory assay, how did you change the level of adhesiveness, most taking into account that IDDSI 4 and British C should not be sticky?

- Should I understand that adhesiveness values were obtained by and instrumental texture assay? Please, describe the methodology used.

- Figure 1: under which criteria did you decide to stablish the first category for adhesiveness between 0 and 4 mJ? Isn´t it too narrow taking into account that the drop for “hard to swallow” occurs around 8 mJ (similarly to the “easiness” scale which grows rapidly from around that value on).

- Some of foodstuffs are mentioned as representative of different adhesiveness levels (mayonnaise, mashed boiled pumpkin and peanut butter) but it is not indicated to which level were allocated. This information should be given for all the samples tested.

- Were the 30 participants of the VFSS assay the same ones as in the sensory assessments? Please, clarify in the text.

- “The two physicians reached a consensus regarding the presence or absence of food remnants”. Please explain the technical aspects of the criteria stablished in the consensus. Was there a qualitative description for “remnants” which could be explained?

Author Response

We will now begin our response to your review. Above all, we would like to thank you for reviewing our paper so carefully.

INTRODUCTION :

Lacks background on adhesiveness

  • Thank you for your observation. Enhancing the explanation of adhesiveness will likely make the overall understanding of the manuscript easier. We have added a description of adhesiveness at the end of page 1.

MATERIALS AND METHODS :

- Express age values as mean ± SD in materials and methods

  • We have made the revisions. Thanks to your advice, information about the research subjects can now be provided more accurately.

- The sensory test Fork test 1 and IDDSI level 4 and British Classification C are treated as equivalent or similar regarding viscosity. However, Fork test 1 refers to thickened liquids which don´t drip from the fork. On the contrary, IDDSI level 4 and British C level refer to pureed smooth food. It is noteworthy that IDDSI level 4 is also described as “not being sticky” in the DDSI Spoon Tilt Test. This particularity is not described in the Fork test by Park J-W and should be taken into account. On the other hand, Texture C by the British classification refers to thick purée and it can be explained by the following characteristic: “the prongs of a fork make a clear pattern on the surface” (which it might not happen in cat 1 of the fork test), together with the fact that it should not be sticky, as in IDDSI level 4.

I would say that it is too general to describe the range of viscosities tested this way, just mentioning classification categories, and that it is necessary to clearly report which were the foodstuffs used and which were their viscosity values. Sometimes it might be required that author also explained how foodstuffs were prepared.

  • I understand your questions regarding the IDDSI and british level settings. It was indeed an aspect we deliberated upon. Initially, we focused on classifying the stages based on flow tests related to adhesiveness. After all, what we wanted to determine was how such stickiness of the foods could be associated with adhesiveness. The nine foods used in the previous sensory test were all pre-made products available for purchase in general markets, and the three types of food used in this test were all the same products as those used in the previous experiment.

For example, the "Creamy peanut butter" we used, manufactured by Yantai Xufeng Food Company, was chosen because it is one of the most fluid and least sticky among commercially available peanut butters—a product that exhibits flow when lifted with a spoon or fork. I fully acknowledge the feedback regarding the potential ambiguity in categorization. However, as I previously explained, the distinction was made to focus on adhesiveness. Should improvements still be necessary, I am willing to retain only the classification related to the fork test and remove the other references.

- Please, describe in details of the design and experimental conditions of the sensory test performed to define levels of adhesiveness.

In the sensory assay, how did you change the level of adhesiveness, most taking into account that IDDSI 4 and British C should not be sticky?

- Should I understand that adhesiveness values were obtained by and instrumental texture assay? Please, describe the methodology used.

  • We are grateful for the careful attention you have given to our manuscript. The sensory test was conducted based on a precedent study, a copy of which will be attached for your reference; reading it may clarify the flow of our research. In that study, a CT3 texture analyzer was utilized for texture profile analysis, and adhesiveness (mJ) was measured. Given that the sensory test was already conducted in previous research, detailing the related methodology extensively in this paper has its limitations. Even though, we have mentioned the basic details in the 'Determination of Adhesiveness Level' section of our methods. The three foods tested in the current study were prepared identically to those in the sensory test. Additionally, regarding IDDSI 4 and British C, I believe we have already explained our position in response to your earlier comments. We hope our attached manuscript proves helpful.

- Figure 1: under which criteria did you decide to stablish the first category for adhesiveness between 0 and 4 mJ? Isn´t it too narrow taking into account that the drop for “hard to swallow” occurs around 8 mJ (similarly to the “easiness” scale which grows rapidly from around that value on).

à While we acknowledge the reviewer's observation regarding the notable change in the graph around 8 mJ, our decision to establish 4 mJ as a threshold remains valid. This threshold was derived from our preliminary sensory tests, which demonstrated a consistent and clear transition point at which the majority of participants reported the food as 'easy to swallow.' Although the increase in difficulty is evident at 8 mJ, the 4 mJ mark serves as an early indicator of change in the swallowing ease, which is critical for our study's focus on the early detection of swallowing difficulties. We believe that maintaining this demarcation provides a more sensitive measure to detect subtle changes in adhesiveness that have a significant impact on swallowing ease

- Some of foodstuffs are mentioned as representative of different adhesiveness levels (mayonnaise, mashed boiled pumpkin and peanut butter) but it is not indicated to which level were allocated. This information should be given for all the samples tested.

  • Thank you for your detailed review. Based on the sensory test conducted with nine foods in previous research, we divided the stages into three levels, designating mayonnaise as Level 1, mashed boiled pumpkin as Level 2, and peanut butter as Level 3. This information is included in the Results section.

- Were the 30 participants of the VFSS assay the same ones as in the sensory assessments? Please, clarify in the text.

  • The basic setup of the experiment was the same as the previous sensory test, but the participants were different individuals. This has been specified in the VFSS section of the Methods. Thank you.

- “The two physicians reached a consensus regarding the presence or absence of food remnants”. Please explain the technical aspects of the criteria stablished in the consensus. Was there a qualitative description for “remnants” which could be explained?

  • Thank you for your meticulous feedback. The evaluation of the presence or absence of food remnants was conducted solely within the oropharyngeal cavity, which can be discerned visually. Typically, when remnants are found in the pyriform sinuses or vallecular sinuses, they are quantified using a grading system of grade 1, 2, or 3. However, our experiment focused on a binary outcome (present or absent), and therefore, we did not include such a detailed grading system in our manuscript.

Round 2

Reviewer 1 Report

Comments and Suggestions for Authors

The additions certainly help with my understanding of the study. Now it just remains to add a clear statement in the methods at 'determination of adhesiveness levels' that the adhesiveness data is taken from ref 8 (thank you for attaching to the review) so it is clear you haven't re-measured it in this study. Also in VFSS methods cite the appropriate reference at 'previous sensory tests'. 

Comments on the Quality of English Language

A brief editorial check on the English would help, particularly in those sections that have been added at the previous review stage.

Author Response

The additions certainly help with my understanding of the study. Now it just remains to add a clear statement in the methods at 'determination of adhesiveness levels' that the adhesiveness data is taken from ref 8 (thank you for attaching to the review) so it is clear you haven't re-measured it in this study. Also in VFSS methods cite the appropriate reference at 'previous sensory tests'. 

  • Your guidance has made our manuscript more constructive, and we sincerely appreciate you taking the time to review the revisions. We have added references to the 'Determination of adhesiveness level' and 'VFSS' sections of the Methods to ensure a clear distinction is made regarding content conducted in previous studies. Once again, we truly thank you.

Reviewer 2 Report

Comments and Suggestions for Authors

I appreciate the effort to modify the manuscript. In my opinion some aspects are clearer now. I understand that space limiations are the frame to which the text must be adapted and thus, explanations about previously described procedures cannot take too long. However, I still miss some explanations which might be relevant to understand this text as a whole. Although it is understandable that extense details for previous methods/preocedures are not given, brief references would help to understand the manuscript by itself.

As the most noticeable issues I would remark the following ones:

- In the section “Determination of adhesiveness level”→ I find that the text in this section contents a mix between both, the instrumental assay and the sensory evaluation of the swallowing ease. I suggest to organize this section more clearly.

- Moreover, sensory assay to evaluate swallowing ease is not enough described, yet. Information is not given about evaluations settlement, conditions, presentation of the samples, scales, order, repetitions...It could be an option to refer to a previous paper, if that´s suitable.

- VFSS → I was asking for criteria to identify remmants and authors give a very clear explanation to this questions within their response which, I suggest, should be included briefly in the text, for a better comprehension by those not familiarised with VFSS.

There are some minor aspects which should be modified as well:

- Authors refere to the assay with the texture analyser as “sensory assay”. In order to avoid missunderstanding I suggest that this assay is named “intrumental texture/sensory assay”.

- Brands for the foodstuffs used in the assay should be indicated, as you have explained in your response for the peanut butter. This is a good example of the need to give details about the products, as it is especifically mentioned that the least sticky one was chosen. As an alternative, if there is a previous paper in which these same products were used and presented in detail, it should be referred to clearly in this material and methods section.

- Space needed between: “6.5years”, parenthesis in L1 at section “Determination of adhesiveness level”

- Results, L4 → “resoponses, We...”

Author Response

Letters to reviewer

Your guidance has made our manuscript more constructive, and we sincerely appreciate you taking the time to review the revisions.

In the section “Determination of adhesiveness level”→ I find that the text in this section contents a mix between both, the instrumental assay and the sensory evaluation of the swallowing ease. I suggest to organize this section more clearly.

à First of all, thank you for your meticulous review. While re-examining the paper, I thought that the flow of the paragraphs could be clearer, and your advice was helpful in that regard. Therefore, I have made significant revisions to the basic paragraphs to ensure that the instrumental assay and sensory evaluation are well organized and distinct. Thank you.

Moreover, sensory assay to evaluate swallowing ease is not enough described, yet. Information is not given about evaluations settlement, conditions, presentation of the samples, scales, order, repetitions...It could be an option to refer to a previous paper, if that´s suitable.

à Thank you for reviewing our manuscript. We have revised the 'Determination of adhesiveness level' section, ensuring it clearly indicates that we used the results from sensory tests conducted in previous research by adding the appropriate references. These references should allow for a detailed understanding of the sensory assay involved.

VFSS → I was asking for criteria to identify remmants and authors give a very clear explanation to this questions within their response which, I suggest, should be included briefly in the text, for a better comprehension by those not familiarised with VFSS.

à Thank you for bringing this to our attention. We have supplemented the manuscript with additional information regarding remnants. The rationale behind focusing on the presence or absence of remnants now seems much more coherent. Thank you.

There are some minor aspects which should be modified as well:

Authors refere to the assay with the texture analyser as “sensory assay”. In order to avoid missunderstanding I suggest that this assay is named “intrumental texture/sensory assay”.

à Thank you for your initial mention of the need to clearly separate the instrumental assay and the sensory evaluation of swallowing ease. We have created new sections in the manuscript to address this, and we believe there will be no misunderstanding regarding the instrumental texture/sensory assay. Thank you.

Brands for the foodstuffs used in the assay should be indicated, as you have explained in your response for the peanut butter. This is a good example of the need to give details about the products, as it is especifically mentioned that the least sticky one was chosen. As an alternative, if there is a previous paper in which these same products were used and presented in detail, it should be referred to clearly in this material and methods section.

à Thank you for your thorough review. In this study, the representative foods for each level, such as mayonnaise, boiled pumpkin, and peanut butter, are the same as those used in previous research. We used peanut butter manufactured by Yantai Xufeng Food Company again, and for mayonnaise, we used the most commonly available brand in Korean markets, made by Ottogi. The pumpkin was prepared in the same way as in our previous study. However, there are quantitative limitations to describing each food's brand in detail. Instead, the sentence in the methods section of the VFSS, [This protocol was conducted in accordance with the sensory test protocols outlined in our previous study], could be further elaborated.

Space needed between: “6.5years”, parenthesis in L1 at section “Determination of adhesiveness level”

Results, L4 → “resoponses, We...”

à Thank you for checking so meticulously. I have made the corrections to the points you highlighted.